# Co-Delivery of the Human NY-ESO-1 Tumor-Associated Antigen and Alpha-GalactosylCeramide by Filamentous Bacteriophages Strongly Enhances the Expansion of Tumor-Specific CD8+ T Cells

**DOI:** 10.3390/v15030672

**Published:** 2023-03-02

**Authors:** Roberta Manco, Luciana D’Apice, Maria Trovato, Lucia Lione, Erika Salvatori, Eleonora Pinto, Mirco Compagnone, Luigi Aurisicchio, Piergiuseppe De Berardinis, Rossella Sartorius

**Affiliations:** 1Institute of Biochemistry and Cell Biology (IBBC), National Research Council (CNR), 80131 Naples, Italy; 2Takis Biotech, 00128 Rome, Italy; 3Neomatrix Biotech, 00128 Rome, Italy

**Keywords:** filamentous bacteriophage, NY-ESO-1, iNKT, CD8+ T cell, alpha-GalactosylCeramide, vaccine

## Abstract

Tumor-associated antigens (TAAs) represent attractive targets in the development of anti-cancer vaccines. The filamentous bacteriophage is a safe and versatile delivery nanosystem, and recombinant bacteriophages expressing TAA-derived peptides at a high density on the viral coat proteins improve TAA immunogenicity, triggering effective in vivo anti-tumor responses. To enhance the efficacy of the bacteriophage as an anti-tumor vaccine, we designed and generated phage particles expressing a CD8+ peptide derived from the human cancer germline antigen NY-ESO-1 decorated with the immunologically active lipid alpha-GalactosylCeramide (α-GalCer), a potent activator of invariant natural killer T (iNKT) cells. The immune response to phage expressing the human TAA NY-ESO-1 and delivering α-GalCer, namely fdNY-ESO-1/α-GalCer, was analyzed either in vitro or in vivo, using an HLA-A2 transgenic mouse model (HHK). By using NY-ESO-1-specific TCR-engineered T cells and iNKT hybridoma cells, we observed the efficacy of the fdNY-ESO-1/α-GalCer co-delivery strategy at inducing activation of both the cell subsets. Moreover, in vivo administration of fdNY-ESO-1 decorated with α-GalCer lipid in the absence of adjuvants strongly enhances the expansion of NY-ESO-1-specific CD8+ T cells in HHK mice. In conclusion, the filamentous bacteriophage delivering TAA-derived peptides and the α-GalCer lipid may represent a novel and promising anti-tumor vaccination strategy.

## 1. Introduction

Cancer immunotherapy aims at harnessing the host immune system against selected tumor antigens, using cancer vaccines, or amplifying pre-existing anti-tumor immune responses through the administration of immunostimulating molecules [1].

Tumor-associated antigens (TAAs) and tumor-specific antigens (TSAs) represent promising targets for anti-cancer vaccines [2]. TAAs are autologous proteins over-expressed in many types of malignant tumors [3]. Carcino-embryonic and melanoma-associated TAAs have been widely used as biomarkers for a variety of cancers and have been employed in a huge number of clinical studies as components of different peptide-based vaccines [4]. However, the engineering of optimal delivery systems, the incorporation of adjuvants, and the strategies to overcome self-tolerance are still needed to improve the efficiency of peptide-based cancer vaccines [5].

NY-ESO-1 belongs to the cancer testis 6 antigen (CT6) family encoded by the *CTAG1B* gene, and it is not expressed in normal healthy tissues. The expression of NY-ESO-1 in cancers is a well-documented phenomenon with studies reporting that approximately 75% of cancer patients express this antigen at some stage during the course of their illness [6]. It represents a promising target for immunotherapy due to its high and frequent expression in malignant tumors and its ability to elicit powerful humoral and cellular responses. Indeed, vaccination with NY-ESO-1-based formulations administered alone or in combination with adjuvants [7,8] or checkpoint inhibitors [9] induced both humoral and cellular anti-tumor responses in clinical trials; thus, the NY-ESO-1 antigen is often included in multi-epitope cancer vaccines [10].

Alpha-GalactosylCeramide (α-GalCer) represents a novel immunotherapeutic agent with strong effects as a vaccine adjuvant [11]. α-GalCer is recognized by invariant natural killer T (iNKT) cells, a subpopulation of T cells involved in the innate immune response. iNKTs recognize lipid antigens presented by CD1d molecules expressed by antigen-presenting cells (APCs), such as dendritic cells (DCs), by means of their semi-invariant TCR composed via the invariant TCRα (Vα14-Jα18 in mice and Vα24-Jα18 in humans), paired with a restricted repertoire of β chains, including Vβ2, Vβ7, and most commonly Vβ8.2 in mice and the Vβ11 TCRβ chain in humans [12,13]. iNKT cells have potent immunomodulatory activity, given their ability to secrete Th1 and Th2 cytokines, and their activity is associated with enhanced protection against tumors [14,15] and pathogens [16,17,18]. However, despite the wide use of soluble α-GalCer in preclinical and clinical studies, its effectiveness is still uncertain and more innovative approaches for enhancing the anti-tumor function of α-GalCer are currently being studied [14]. For instance, the chemical conjugation of α-GalCer to antigenic peptides has shown the induction of efficient stimulatory activity by iNKT cells, with the enhancement of cytotoxic T cell responses and therapeutic anti-tumor effects [19,20].

Filamentous bacteriophages for their size, ability to penetrate blood vessels, and easiness of being engineered for the expression of high-density antigens on their surface, can be assimilated to natural nanoparticles with intrinsic adjuvant properties [21,22,23,24]. Moreover, due to their ability to infect and replicate exclusively in bacterial host cells, they are generally considered to be safe for humans [25]. The bacteriophage genome sequence can be modified through the insertion of exogenous sequences, allowing for the expression of an antigen on the phage surface in a high copy number, as N-terminal fusion with the phage structural proteins. We previously demonstrated that therapeutic vaccination using bacteriophages expressing the model antigen OVA_257–264_ and bound to α-GalCer is able to delay the growth of OVA-expressing tumors in animal models [26]. The phage-based vaccine simultaneously delivering α-GalCer and antigenic peptides offers many advantages compared to α-GalCer-peptide chemical conjugates: phages possess an intrinsic adjuvanticity that allows for their use in the absence of exogenous adjuvants, promoting DC maturation and potentially resulting in synergistic effects. In addition, the expression of antigenic peptides in multiple copies on a single phage particle permits the strong multivalent binding of peptides to the APCs, increases the peptide stability, and optimizes the biodistribution and pharmacokinetics [27,28,29]. Moreover, α-GalCer bacteriophages can be further modified to generate multivalent phages exposing different types of peptides [30], or offering the possibility of targeted delivery to a specific tissue or cell subsets by co-displaying antibody fragments or receptor ligands on the phage surface, maximizing the specific uptake [23,31,32].

Here, we engineered and used phage-based nanoparticles co-delivering the human NY-ESO-1 antigen and the α-GalCer lipid to obtain dual-functioning phages able to simultaneously activate iNKT and cytotoxic T cell responses. Via the immunization of HHK mice expressing the human HLA-A2*0201 restriction molecule, we show the in vivo expansion of tumor-specific CD8+ T cells. Thus, the phage nanoparticles have the potential of harnessing innate and adaptive immunity against NY-ESO-1-expressing malignancies.

## 2. Materials and Methods

### 2.1. Cell Lines and Reagents

J76 NFAT-GFP cells were a kind gift from Dr. Acuto [33] and were grown in RPMI-1640 (GIBCO, Billings, MT, USA) supplemented with 10% fetal bovine serum (FBS), 100 U/mL penicillin G, and 100 μg/mL streptomycin.

CD1d-restricted Vα14i NKT hybridoma FF13 [34] cells were a kind gift from Dr. De Libero (Department of Biomedicine, University of Basel and University Hospital Basel, Basel, Switzerland) and were cultured in RPMI 1640 (GIBCO) medium supplemented with 100 U/mL penicillin G, 100 μg/mL streptomycin, and 10% FBS (GIBCO).

The synthetic peptide NY-ESO-1_157–165_ (Val165) (SLLMWITQV) was from GenScript Biotech (The Netherlands). PE-HLA-A2*0201/SLLMWITQV MHC dextramers were from Immudex (Copenhagen, Denmark).

Synthetic α-GalCer (KRN7000) (2S,3S,4R)-1-O-(α-D-galactosyl)-N-hexacosanoyl-2-amino-1,3,4-octadecanetriol (BML-SL232-1000) was purchased from Vinci Biochem.

### 2.2. Mice

C57BL/6 mice were purchased from Charles River (Lecco, Italy) and housed at the IGB “A. Buzzati-Traverso” Animal House Facility (Naples, Italy) under standard pathogen-free conditions abiding by institutional guidelines. Transgenic HHK mice, expressing the human β2-microglobulin and HLA-A2*0201 with the mouse H-2Kb α3 transmembrane and cytoplasmic domains [35], were housed at the Plaisant animal house facility (Rome, Italy), according to the national legislation, and kept in standard conditions in accordance with Takis’ ethics committee approval.

### 2.3. Phage Nanoparticle Construction and Production

Recombinant hybrid bacteriophage fdNY-ESO-1-expressing CTAG1B_157–165_ (Val165) SLLMWITQV -pVIII fusion proteins were generated according to the protocol described in [36]. Briefly, complementary DNA oligos encoding the NY-ESO-1_157–165_ (Val165) HLA-A2-restricted peptide (5′-GGAGGGTagcctgctgatgtggattacccaggtgGACGATCCCGC-3′; 5′-CTTGGCGGGATCGTCcacctgggtaatccacatcagcaggctACCCTCCGC-3′) were annealed and cloned into SacII-StyI-digested fdAMPLAY88 phage genomes [37] containing two copies of the pVIII protein-encoding sequence, one wild-type and one harboring the SacII-StyI restriction sites. This second copy expression is regulated by the isopropyl-beta–d-thiogalactopyranoside (IPTG)-inducible promoter pTac, and the resulting hybrid phage progeny expresses recombinant copies of the major coat protein pVIII interspersed with the wild-type pVIII copies on the coat surface of each single virion. fdAMPLAY88 wild-type (fdWT) and fdNY-ESO-1 hybrid filamentous bacteriophages were purified from the supernatant of *E. coli* TG1recO cells previously transformed with the phage DNA, according to [37]. Briefly, bacteria containing the phage DNA were grown in 2XTY medium (16 g/L Tryptone, 10 g/L yeast extract, 5.0 g/L NaCl) for 16–18 h in the presence of 100 μg/mL ampicillin. The addition of 0.1 mM IPTG (Sigma-Aldrich, Milan, Italy) to the growing cultures (at absorbance A600 = 0.25 optical density (OD)) drove the expression of the recombinant NY-ESO-1-pVIII proteins. Phages were precipitated from the supernatant of *E. coli* cultures by adding 20% polyethylene glycol 6000 (PEG, Sigma-Aldrich) and 2.5 M NaCl (Sigma-Aldrich) to the supernatant. Phages were then harvested via centrifugation (16,000× *g*), the pellet was resuspended in 10 mM Tris/1 mM EDTA pH 8.0 buffer (TE), and the virions were purified via ultracentrifugation (240,500× *g*) on cesium chloride gradient (0.5 mg/mL; Sigma-Aldrich). The particles were then dialyzed in phosphate-buffered saline (PBS) 1× and the concentration of the virions was determined using a spectrophotometer.

The number of copies of pVIII displaying the NY-ESO-1 peptide on the capsid of the hybrid bacteriophages fdNY-ESO-1 was estimated based on the relative yields of the various N-terminal sequence analysis of the purified phage particles, which resulted in about 15% of recombinant pVIII for each preparation.

### 2.4. Alpha-GalactosylCeramide Decoration of Phage Particles

The elimination of lipopolysaccharides (LPS) from phage particles was performed via Triton X-114 (Sigma-Aldrich) extraction. Briefly, Triton X-114 was added to the phage preparations to a final concentration of 1% *v/v* by vigorous vortexing. The solution was incubated at 4 °C for 5 min, then for 5 min at 50 °C, and centrifuged (20,000× *g*) for 10 min at 25 °C. The upper phase containing the virions was recovered and subjected to Triton X-114 phase separation for 8–10 cycles. Finally, phage particles were subjected to cesium chloride gradient centrifugation, dialyzed against PBS 1×, and assayed for LPS content using the Limulus Amebocyte Lysate assay (Pierce™ Chromogenic Endotoxin Quant Kit, Thermo Scientific, Waltham, MA, USA), according to the manufacturer’s instructions.

Bacteriophages and the α-GalCer synthetic analogous KRN7000 in dimethyl sulfoxide (Sigma-Aldrich) were combined in a 10:1 ratio (μg phages: μg α-GalCer) and stirred at 4 °C overnight. α-GalCer-virions were subjected to cesium chloride gradient ultracentrifugation and dialyzed against PBS 1×.

The presence of α-GalCer in the phage preparations was analyzed via the UPLC-MS/MS method as reported in Sartorius et al. [26], and confirmed by the in vitro biological assay as described below.

### 2.5. Viral Transduction of Jurkat Cells

Recombinant lentiviral particles were produced as described in [38]. Briefly, HEK293T cells (ATCC CRL 1573) were transfected with 8 μg of pLENTI-1G4 plasmid, encoding the 1G4 T cell receptor (TCR) (NY-ESO-1-specific alpha and beta chains), 4 μg of the pCMVΔR8.2 packaging plasmid, and 4 μg of the pMD2.G vesicular stomatitis virus G glycoprotein (VSV-G) envelope-expressing plasmid. The cell culture supernatant was harvested 72 h after transfection, filtered using 0.45 μM PES filter units, and used for J76 NFAT-GFP cell infection by culturing 500,000 cells in 10 mL of collected cell culture supernatant containing the lentiviral particles.

Forty-eight h later, cells were harvested and washed with ice-cold FACS staining buffer (PBS 1×, 2% FBS), then stained with APC-conjugated anti-human CD3 (clone HIT3A, Sony), washed twice with FACS staining buffer, and sorted using the FACSAria™ (BD) cell sorter. Sorted cells were amplified, assayed for CD3 expression, and used for subsequent experiments.

### 2.6. Human and Mouse Dendritic Cell Generation

Mouse bone-marrow-derived DCs (BM-DCs) were derived from precursors isolated from the tibiae of C57BL/6 mice and cultured in RPMI-1640 media supplemented with 10% FBS, 5 μM 2-Mercaptoethanol, 1 mM Glutamine, and 1 mM Sodium Pyruvate in the presence of 200 U/mL recombinant murine granulocyte/macrophage colony-stimulating factor (GM-CSF, Peprotech, NJ, USA). Immature DCs were collected on day seven of culture and were assayed for the DC phenotype via FACS analysis after staining with the monoclonal antibody PE-Cy7 anti-CD11c (HL3, BD Biosciences, San Jose, CA, USA).

Human DCs were generated from human peripheral blood mononuclear cells (PBMCs). Human peripheral blood was obtained from discharge materials left during the manufacturing to prepare blood bags to use for medical purposes and no identifying information on the donor was retained. The samples were recovered from different transfusion centers in the Campania region (Italy), blood donors were aware that leftover material could be occasionally used for various research purposes, and they routinely gave their informed consent for this use when donating blood. PBMCs were isolated via Ficoll density gradient centrifugation. HLA-A2 expression was verified via FACS analysis using FITC anti-human HLA-A2 antibody (BB7.2, Biolabs) and the FACSCanto™ (BD).

Monocytes were isolated using anti-CD14 antibody-conjugated magnetic beads (Miltenyi, Koto, Japan) and differentiated into DCs by culturing cells in complete RPMI-1640 media supplemented with 5% autologous plasma, 1 mM glutamine, 1 mM sodium pyruvate, human recombinant IL-4 (250 U/mL), and human recombinant GM-CSF (1000 U/mL) (Peprotech, Cranbury, NJ, USA) for 5 days. Cultured cells were further fed on day three with complete RPMI-1640 media supplemented with the same concentration of cytokines. Cells were assayed for DC phenotype via FACS analysis after staining with FITC anti-human CD14 and PE anti-human CD1c.

### 2.7. CD83 Expression on Human DCs and NFAT Assay

Human monocyte-derived DCs (50,000) were pulsed with fdNY-ESO-1, fdNY-ESO-1/α-GalCer, fdNY-ESO-1 plus soluble free α-GalCer, fdWT, or with synthetic NY-ESO-1 peptide (pNY-ESO-1) in 200 μL of complete RPMI-1640 medium for 18 h. The expression of the costimulatory marker CD83 was evaluated on DCs by staining with FITC anti-human CD83 antibody (Biolegend, San Diego, CA, USA). DCs were plated at a density of 50,000/well and then 50,000 1G4 TCR-transduced J76 NFAT-GFP cells were added to the culture. Cells were harvested 24 h later and stained with PE anti-human CD1c (Biolegend) and APC anti-human CD3. The percentage of GFP-expressing cells was evaluated via FACS analysis (FACSCanto™, BD) on CD3+ gated cells; CD1c+-high DCs were excluded from FACS analysis. At least 30,000 events were collected for each cell culture condition.

### 2.8. In Vitro iNKT Response

Mouse BM-DCs (100,000/well) or human DCs (25,000/well) were incubated in RPMI-1640 medium supplemented with 10% FBS, 5 μM 2-Mercaptoethanol, 1 mM glutamine, and 1 mM sodium pyruvate for 2 h with different concentrations of free α-GalCer (1.5, 15, 150 ng/mL), fdNY-ESO-1, or fdNY-ESO-1/α-GalCer bacteriophages (0.5, 5, and 50 μg/mL, containing the above-mentioned amounts of α-GalCer, respectively). In some experiments, cells were also incubated with fdNY-ESO-1 mixed with soluble α-GalCer. After the incubation, cells were washed and co-cultured with mouse iNKT cells (50,000/well) for 40 h. IL-2 released into co-culture supernatants was measured via ELISA. Supernatants (0.1 mL/well) were assayed in duplicate using mouse IL-2 ELISA MAX™ Standard (Biolegend) according to the manufacturer’s instructions.

### 2.9. In Vivo Immunization and Dextramer Staining

HHK mice (*n* = 4) were primed (Day 0) via subcutaneous injection of 80 μg of fdNY-ESO-1, either delivering α-GalCer in PBS 1× or not. The administration of PBS 1× alone or 50 μg of NY-ESO-1_157–165_ synthetic peptide with 50 μg of CpG in incomplete Freund’s adjuvant (IFA, Sigma) was performed and used as the negative and positive controls, respectively. Boosters (Days 14 and 21) were performed under the same conditions. After 4 weeks, mice were sacrificed and splenocytes were harvested. Cells were resuspended in complete RPMI-1640 medium after erythrocytes lysis with 4 mL ACK (Ammonium-Chloride-Potassium) Lysing Buffer (Life Technologies, Carlsbad, CA, USA) for 10 min at room temperature, and then HLA-A2*0201/SLLMWITQV MHC dextramer staining was performed.

Briefly, 2,000,000 splenocytes were cultured in 96-well plates and washed with ice-cold PBS 1× (300× *g* for 5 min). Dead cells were excluded using the Horizon Fixable Viability Stain 575V (BD Horizon). Splenocytes were washed twice with FACS staining buffer (PBS 1×, EDTA 2 mM, FBS 0.5%) and incubated with anti-Fcγ receptor (Mouse BD Fc Block™) followed by surface staining with AlexaFluor488 anti-CD3e, PerCP-eFluor710 anti-CD4, APC-eFluor780 anti-CD8 (Thermofisher, Waltham, MA, USA), and PE-HLA-A2*0201/SLLMWITQV MHC dextramers for 20 min at 4° in the dark. The antibody mix containing Abs specific for the cell surface markers without dextramers was used as a negative control. Following surface staining, samples were washed twice with FACS staining buffer and analyzed using the CytoFLEX flow cytometer (Beckman Coulter, Brea, CA, USA) and FlowJo software.

## 3. Results

### 3.1. Production of NY-ESO-1 Bacteriophage Bound to α-GalCer

To obtain fdNY-ESO-1 bacteriophages, whose capsids contain modified major coat proteins (pVIII) displaying NY-ESO-1-derived peptide interspersed with the wild-type pVIII, the SLLMWITQV peptide was added at the N-terminal end of pVIII via exogenous sequence cloning into the phage genome. N-terminal sequence analysis proved that about 15% of the 2700 pVIII coat proteins per virion displayed the SLLMWITQV peptide (equal to about 3 μg of peptide/100 μg of phages). LPS-free bacteriophages were obtained via endotoxin purification with Triton X-114 treatment, resulting in an LPS residual contamination < 0.5 EU/mL, and then bound to the synthetic α-GalCer glycolipid in LPS-free conditions.

### 3.2. Antigen-Specific T Cell Response to Bacteriophages Carrying Tumor-Associated Antigens

The efficiency of the NY-ESO-1 peptide delivery via a bacteriophage-based nanocarrier at inducing a specific CD8+ T cell response was evaluated by measuring the antigen recognition and consequent activation of NY-ESO-1 antigen-specific T cells.

For this purpose, a human lymphocyte cell line was engineered for the expression of the NY-ESO-1_157–165_-specific 1G4 TCR, a widely used optimized TCR with high intra-chain affinity [39]. The expression of 1G4 TCR on the surface of the J76 NFAT-GFP T lymphocyte line (endowed with NFAT-dependent GFP expression) was evaluated via FACS staining with the anti-human CD3 antibody. In order to have a homogeneous cell population, CD3 positive cells were sorted and expanded in culture, leading to a population with 94.1% of the CD3 expression rate (Appendix A).

Assays were then performed to test the ability of human DCs to present the NY-ESO-1 antigen exposed on filamentous bacteriophages to T cells. Monocyte-derived DCs from HLA-A2.1+ healthy donors were incubated with fdNY-ESO-1, fdNY-ESO-1 mixed with soluble α-GalCer (fdNY-ESO-1+α-GalCer), or bacteriophages simultaneously co-delivering the NY-ESO-1 epitope fused to pVIII and the α-GalCer lipid (fdNY-ESO-1/α-GalCer). The synthetic peptide NY-ESO-1 (pNY-ESO-1) and the wild-type bacteriophage (fdWT) were used as a positive and negative control, respectively. DCs were then co-cultured with 1G4 J76 NFAT-GFP cells and antigen-specific T cell activation was evaluated via GFP reporter gene expression. DCs (CD1c+ high) were excluded from the analysis.

The fdNY-ESO-1/α-GalCer bacteriophage is properly internalized and presented by human DCs (Figure 1A,B), inducing the activation of the engineered antigen-specific T cells. Bacteriophage-mediated T cell activation was greater than the one induced by synthetic peptide alone and the one obtained using fdNY-ESO-1 without the lipid. Moreover, this response was also significantly higher than the administration of fdNY-ESO-1 mixed with free α-GalCer, demonstrating the efficacy of the fdNY-ESO-1/α-GalCer co-delivery strategy.

### 3.3. iNKT Response to Phage Bound to α-GalCer

The immune-stimulatory activity of the α-GalCer lipid in phage/lipid preparations was determined by measuring cytokine production via a mouse iNKT hybridoma [34]. Both murine and human DCs were derived from precursors in the presence of the appropriate growth factors, as described in Methods. Murine and human DCs were then incubated with increasing concentrations of fdNY-ESO-1, fdNY-ESO-1/α-GalCer, and free α-GalCer in the same molar amount or fdNY-ESO-1+α-GalCer mixture. Cells were then washed and co-cultured with the iNKT hybridoma FF13, and IL-2 released in the supernatant was measured via ELISA assay.

Either murine or human DCs internalized fdNY-ESO-1/α-GalCer and induced IL-2 production by iNKT cells (Figure 2A,B), indicating that the internalized α-GalCer is exposed in complex with the CD1d molecule. No IL-2 was secreted in cultures pulsed with fdNY-ESO-1 in the absence of α-GalCer (data not shown).

We then sought to evaluate whether co-administration of fdNY-ESO-1 plus soluble α-GalCer was able to induce a response comparable to the one induced by fdNY-ESO-1/α-GalCer. The iNKT response elicited by human DCs pre-pulsed with fdNY-ESO-1/α-GalCer, or with fdNY-ESO-1 plus soluble α-GalCer, was thus analyzed. The soluble α-GalCer was added in the same concentration as the one present in the fdNY-ESO-1/α-GalCer and was estimated using quantitative LC–MS/MS analysis. As reported in Figure 2C, the mix of fdNY-ESO-1+α-GalCer induced a response comparable to the one produced by the soluble α-GalCer alone and was much lower compared to fdNY-ESO-1/α-GalCer. To evaluate the ability of fdNY-ESO-1/α-GalCer to induce DC maturation, the phenotype of pre-pulsed DCs was evaluated via surface staining of the CD83 activation marker. DCs incubated with fdNY-ESO-1/α-GalCer showed a higher expression of CD83 compared to DCs pulsed with fdNY-ESO-1 mixed with the free lipid (Figure 2D).

### 3.4. Co-Delivery of Peptides and α-GalCer Enhances CD8 T Cell Responses In Vivo

Efficiency of fdNY-ESO-1 decorated with α-GalCer to elicit a strong specific immune response in vivo was tested in transgenic HHK mice. In detail, β2-microglobulin negative, Db negative, HLA-A2.1 transgenic HHK mice were injected subcutaneously with fdNY-ESO-1/α-GalCer, fdNY-ESO-1, or with pNY-ESO-1 in the presence of both the TLR9 activator CpG and IFA as the positive control. Two booster injections were administered on Days 14 and 21 (Figure 3A).

The analysis of splenic CD8+ T cells specific for the NY-ESO-1 TAA showed a higher percentage of specific T cells when mice were immunized with fdNY-ESO-1 bound to α-GalCer, compared to fdNY-ESO-1 and pNY-ESO-1 plus CpG and IFA (Figure 3B,C). The response is strictly CD8+-restricted, as proven by the absence of an NY-ESO-1-specific CD4+ T cell response (Appendix A).

These results suggest that the antigen-specific immune response is boosted by iNKT cell stimulation using the fdNY-ESO-1/α-GalCer co-delivery strategy.

## 4. Discussion

NY-ESO-1 is one of the main TAA targets for human cancer therapy. It is expressed only in germ cells and trophoblasts and is upregulated in 20–70% of cancers. Several immunogenic NY-ESO-1 epitopes have been identified covering more than 80% of the European population [39]. When NY-ESO-1_157–165_ peptide was modified with a valine substitution in position 165 (9 V) (SLLMWITQV), a better binding to the HLA-A2 allele was proved, and it was recognized 100 times more efficiently compared to the wild-type peptides, inducing a higher NY-ESO-1-specific CD8+ stimulation in vitro [40]. However, to date, no successful NY-ESO-1 cancer vaccine based on free NY-ESO-1_157–165_ peptide has been reported.

Here, we exploited the filamentous bacteriophage antigen delivery system, improved by the co-delivery of the immunomodulating lipid α-GalCer, to stimulate innate and adaptive responses against the human tumor antigen NY-ESO-1. The filamentous bacteriophage was able to simultaneously deliver antigenic peptides and glycolipids to DCs, inducing high and specific T cell responses in a tumor-bearing mouse model of therapeutic vaccination [26]. As in the mouse model, here we showed that human DCs internalized the bacteriophage and loaded vectorized human TAA-peptide on MHC-I molecules, inducing the proliferation of T cells expressing the high-affinity NY-ESO-1 TCR. Importantly, the administration of fdNY-ESO-1/α-GalCer enhanced the activation of NY-ESO-1-specific T cells.

By using a murine hybridoma we also showed that fdNY-ESO-1/α-GalCer was able to stimulate IL-2 production by iNKT cells. In fact, murine iNKT cells can recognize human CD1d-lipid complexes [41], due to highly conserved sequences between murine and human CD1d proteins [42] and thanks to the homology between murine Vα14 and human Vα24 and murine Vβ8.2 and human Vβ11 chains [43].

Either murine or human DCs gave rise to a more efficient response when the lipid was delivered by the bacteriophage compared to the response induced by the soluble lipid, administered in equimolar concentrations. Indeed, human DCs were found to be unable to activate iNKT cells at low concentrations of the soluble lipid (1.5 ng/mL). This may be due to the lower efficiency of mouse iNKT hybridoma to recognize the human lipid–CD1d complex. In contrast, α-GalCer delivered by the phage was very efficient at inducing iNKT cell activation, demonstrating that the co-delivery of the immunostimulating lipid on the phage carrier significantly increases the immunogenicity of the α-GalCer also when presented by human DCs.

Indeed, the use of synthetic peptides chemically conjugated to immunostimulating lipids, such as a prodrug form of α-GalCer or its derivatives, showed the promising potential of the combined use of iNKT and T cell antigens to increase immune responses in different pathological conditions such as malaria [44], allergies [45], and hematopoietic [46] and solid tumors [20,47]. Moreover, α-GalCer-peptide administration further demonstrated the efficacy of the simultaneous co-delivery of iNKT and T cell antigens compared to the use of free α-GalCer, suggesting the importance of releasing both components within the same antigen-presenting cell, for the iNKT-mediated stimulation of antigen-specific T cells.

Due to the better performance of fdNY-ESO-1/α-GalCer, we hypothesized an enhancement in DC functions. Indeed, we found a high percentage of DCs expressing the CD83 activation marker after exposure to fdNY-ESO-1/α-GalCer, and synthetic analogs of α-GalCer have been found to activate macrophages via TLR4-dependent signaling pathways [48]. Further studies will be needed to elucidate whether the higher T cell activation is due to a synergic activity exerted by α-GalCer on iNKT cells and DC maturation, and/or whether the lipid can directly act on T cells as well.

To investigate antigen-specific CD8+ T cell activation in vivo, we have exploited the transgenic HHK mouse model which provides the possibility to analyze the immune response against human TAAs. Although these animals have a significant reduction in the overall number of peripheral CD8+ T lymphocytes, HHK transgenic mice represent a valuable model for the preclinical studies of the HLA-A2.1-restricted CD8+ T cell response [49]. Moreover, we took advantage of the high homology present between human and murine CD1d molecules to exploit the activation of innate immune responses mediated by the immunostimulating lipid α-GalCer to support and potentiate the T cell functions mediated by the bacteriophage.

We emphasize that in our model the NY-ESO-1 peptide delivered by bacteriophage induces a high expansion of antigen-specific CD8+ T lymphocytes. Although phage administration did not provide exogenous adjuvants, a high amount of specific CD8+ T cells were induced by the NY-ESO-1 TAA vehiculated by the fd bacteriophage.

The filamentous phage has been shown to have intrinsic adjuvant functions by inducing DC maturation in the absence of exogenous adjuvants [21,22,23,24]. We found that bacteriophage-mediated α-GalCer delivery increases the lipid presentation to iNKT cells compared to the α-GalCer administered in soluble form. This provides a higher activation of iNKT responses that can further help DC maturation and antigen presentation and thus potentiate CD8 T cell functions. In agreement with our findings, an alternative vaccine based on PLGA-containing immunogenic peptides and the synthetic α-GalCer analog enhances CD8 and CD4 T cell responses against NY-ESO-1 in vivo [50].

Filamentous bacteriophages are largely employed in nanotechnology. Their length (up to 800 nm) and their nanoparticulate structure allow the creation of templates or scaffolds for the delivery of drugs, antibodies, or antigenic peptides to be used in immunotherapeutic approaches [51,52]. Phage vaccines are considered to be more advantageous than other vaccine approaches because they are more stable for storage and transport and can be produced in large amounts at a low cost. Bacteriophages are unable to replicate in eukaryotic cells and phage administration in humans has shown no adverse effects [27,53,54]. Although there is a need for further studies regarding the phage-activated immune response in the human model, the data obtained in this work suggest that the phage displaying NY-ESO-1_157–165_ peptide and decorated with the immunostimulating lipid α-GalCer could represent an efficient and alternative vaccine model against human cancer.

## 5. Conclusions

The filamentous bacteriophage is an efficient nanocarrier for the simultaneous delivery of antigenic peptides and immunostimulating lipids, such as the α-GalCer, succeeding in activating both the iNKT and antigen-specific T cell responses. The bacteriophage-based nanocarrier has intrinsic adjuvant properties, and the binding to the α-GalCer lipid further enhances the CD8+ T cell activation. In addition, the fdNY-ESO-1/α−GalCer co-delivery strongly enhances both lipid and antigen presentation by DCs, compared to fdNY-ESO-1 mixed with the soluble lipid. The exposure of peptides derived from human TAA on the filamentous phage surface together with the vectorized delivery of an iNKT agonist could represent a novel therapeutic approach for cancer therapy.

## 6. Patents

R.S. and P.D.B. are the inventors of the following patent: “Phage conjugates and uses thereof” (EP3573667).

## Figures and Tables

**Figure 1 viruses-15-00672-f001:**
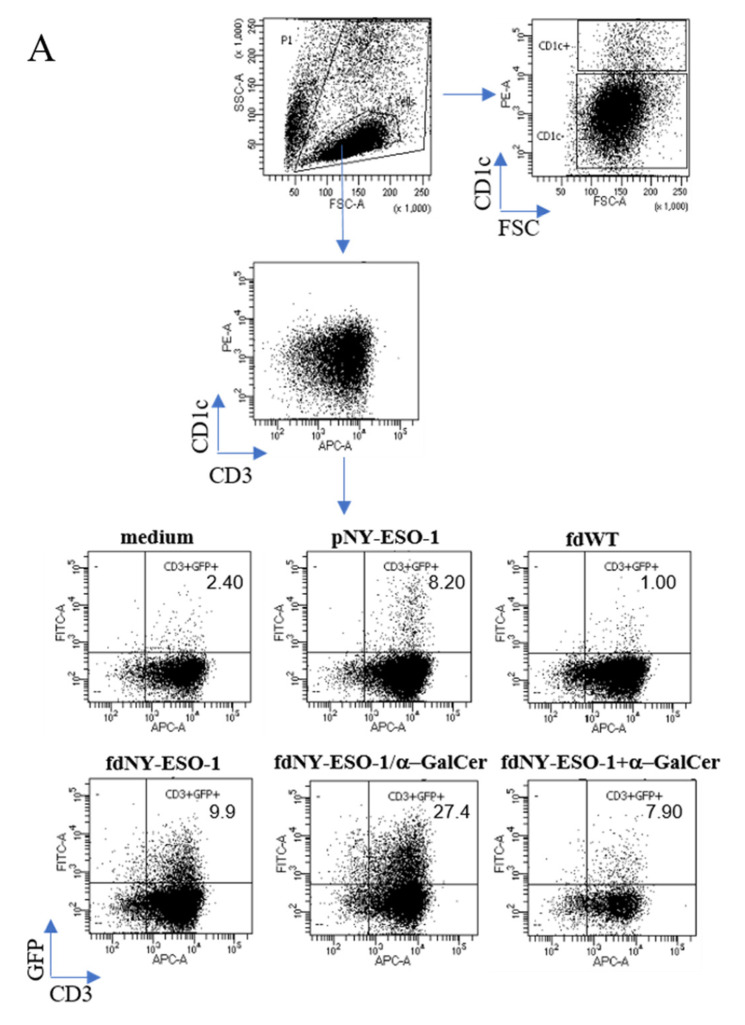
1G4 J76 NFAT-GFP cell response to NY-ESO-1 peptide delivered via phage particles. 1G4 J76 NFAT-GFP cells were co-cultured with human DCs pre-pulsed or not with fdWT, fdNY-ESO-1, fdNY-ESO-1/α-GalCer, fdNY-ESO-1+α-GalCer mixture, or NY-ESO-1 synthetic peptide (pNY-ESO-1). Percentages of GFP+ cells on CD3+ gated cells were measured after 16 h. (**A**) The gating strategy of one representative experiment. DCs were identified as CD1c+ cells, while J76 NFAT-GFP-1G4 T cells were identified as CD1c- cells. T cells were gated as CD1c-/CD3+ cells. GFP+ cells were shown as percentages on CD3+ gated T cells. (**B**) The summary graph displays the mean ± SEM of three experiments. Differences were statistically significant using the two-way ANOVA and Tukey multiple comparisons test (** *p* < 0.01; *** *p* < 0.001).

**Figure 2 viruses-15-00672-f002:**
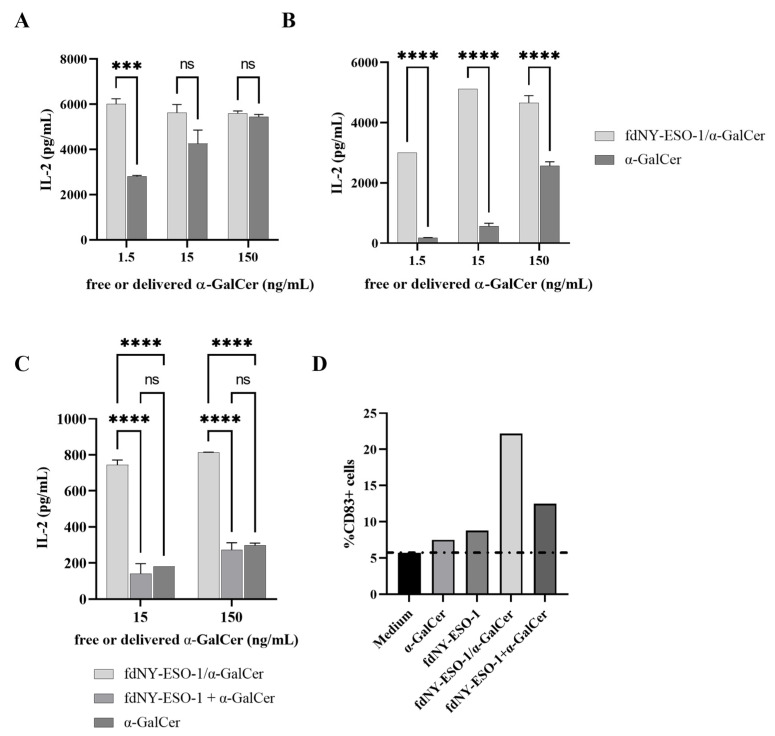
IL-2 production by iNKT cells in response to DCs pre-pulsed with fdNY-ESO-1/α-GalCer. Mouse (**A**) or human (**B**) DCs were pre-pulsed with LPS-free fdNY-ESO-1/α-GalCer or soluble α-GalCer to stimulate FF13 iNKT hybridoma cells. (**C**) iNKT response to human DCs pre-pulsed with fdNY-ESO-1/α-GalCer is compared to the one obtained using fdNY-ESO-1 mixed with soluble α-GalCer or using soluble α-GalCer in equimolar amount. Supernatants were assayed in duplicate after 40 h. Mean ± SEM is reported, and one representative experiment of two is shown. Differences were statistically significant using the two-way ANOVA and Šídák’s or Tukey’s multiple comparisons test (*** *p* < 0.001; **** *p* < 0.0001; ns: not significant). (**D**) Percentages of CD83+ human DCs after pulsing with the different phage particles or soluble α-GalCer. The dotted line represents the percentage of CD83+ cells in unstimulated DC cultures.

**Figure 3 viruses-15-00672-f003:**
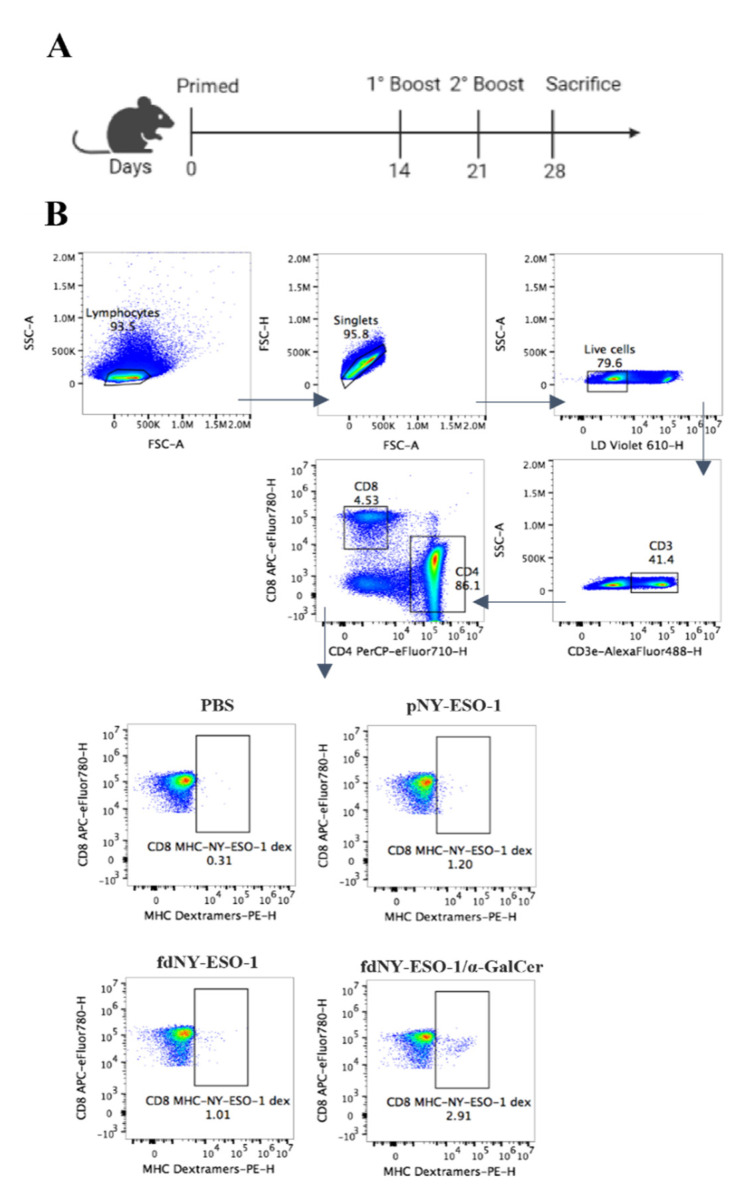
Analysis of CD8+ NY-ESO-1-specific T cells. HHK mice (*n* = 4/group) were subcutaneously injected with fd bacteriophages expressing the NY-ESO-1_157–165_ peptide and decorated or not with α-GalCer, or with NY-ESO-1 synthetic peptide (pNY-ESO-1) in the presence of CpG and incomplete Freund’s adjuvant. Mice were boosted on Days 14 and 21. Seven days after booster injections, mice were sacrificed and the presence of NY-ESO-1-specific CD8+ T cells was evaluated. (**A**) Schematic representation of the experiment. (**B**) FACS analysis gating strategy; one representative sample per group. Lymphocytes and singlets were gated by dimension; live cells were gated as negative for live–dead dye. CD3+ T cells were gated on live cells and MHC-I NY-ESO-1-dextramer positive cells were shown as percentages on CD8+ gated T cells; (**C**) percentages of MHC-I NY-ESO-1-dextramer+/CD8+ cells. Mean ± SEM of *n* = 4 mice/group is reported. Differences were statistically significant using the one-way ANOVA and Dunnet’s multiple comparisons test (* *p* < 0.05).

## Data Availability

Not applicable.

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
