# Peer review of "Co-Delivery of the Human NY-ESO-1 Tumor-Associated Antigen and Alpha-GalactosylCeramide by Filamentous Bacteriophages Strongly Enhances the Expansion of Tumor-Specific CD8+ T Cells"

_viruses, 2023, doi:10.3390/v15030672_

Round 1
Reviewer 1 Report
Manco and coworkers developed a dual-displayed bacteriophage particle that express a cancer-specific peptide antigen (NY-ESO-1) on a second, recombinant major coat protein (p8) and conjugated an immunologically active lipid alpha-GalactosylCeramide to the surface of the bacteriophages. The authors tested the expansion of peptide-specific CD8+ T cells using an in vitro and an in vivo model. Overall, the manuscript is well written and provides a thorough characterization of a novel bacteriophage vaccine platform and its potential utility to the research community.
Author Response
We are deeply grateful to the reviewer for the positive evaluation of our work.Reviewer 2 Report
This manuscript examines effectiveness the filamentous phage fd as a vaccine carrier displaying a cancer-associated antigenic peptide, together with bound aGalCer, an adjuvanting molecule that is normally displayed on DCs and that recruits iNKT cells.
The manuscript is very well written, however there are some serious issues in the terms of results and claims by the authors.
Firstly, nowhere in the manuscript is there a direct (chemical) 1:1 conjugate of antigenic peptide to aGalCer tested, which is a gold standard. The antigen-specific CD8 T-cell response shown here in Figure 3 is minute in comparison to the response to the 1:1 chemical conjugate (numerous publications could have been cited by the authors here).
Secondly, the authors claim that they have conjugated aGalCer to the phage, which normally means that there is chemical conjugation (e.g. https://link.springer.com/protocol/10.1385/1-59259-679-7:255) In this case, the authors claim that binding is conjugation. The term as used here is misleading.
The authors state that the major coat protein g8p of fd is hydrophobic and this is why aGalCer binds to the phage. However, in the phage structure the hydrophobic portion of the g8p is not exposed at the surface (Marvin 2006 JMB and other publications), and the phage virion surface is hydrophilic. The explanation given in the manuscript must be removed as it is factually incorrect.
Authors provide no proof that aGalCer is actually bound to the phage. The authors should, using aGalCer conjugates to fluorescent dyes, show that, when phage are resolved by native phage electrophoresis, the bands are labelled by the dye. Another way to demonstrate association of aGalCer to the phage is preparative agarose gel electrophoresis, extraction and mass spectroscopy.
Author Response
This manuscript examines effectiveness the filamentous phage fd as a vaccine carrier displaying a cancer-associated antigenic peptide, together with bound aGalCer, an adjuvanting molecule that is normally displayed on DCs and that recruits iNKT cells. The manuscript is very well written, however there are some serious issues in the terms of results and claims by the authors.
Firstly, nowhere in the manuscript is there a direct (chemical) 1:1 conjugate of antigenic peptide to aGalCer tested, which is a gold standard. The antigen-specific CD8 T-cell response shown here in Figure 3 is minute in comparison to the response to the 1:1 chemical conjugate (numerous publications could have been cited by the authors here).
We understand the reviewer’s concerns and are thankful for this observation. The use of peptides and proteins chemically conjugated to α-GalCer is certainly a promising technique to potentiate the CD8+ T-cell response, and accordingly, we have discussed the use of α-GalCer -conjugated peptides in the discussion section of the revised version of the manuscript, citing the appropriate literature, as follows (lanes 393-400):
Indeed, the use of synthetic peptides bound to immunostimulating lipids, such as a prodrug form of α-GalCer or its derivatives, showed the promising potential of the combined use of iNKT and T cell antigens to increase immune responses in different pathological conditions such malaria [36] (Meijlink MA, Chua YC, Chan STS, Anderson RJ, Rosen-berg MW, Cozijnsen A, Mollard V, McFadden GI, Draper SL, Holz LE, Hermans IF, Heath WR, Paint-er GF, Compton BJ. 6″-Modifed α-GalCer-peptide conjugate vaccine candidates protect against liver-stage malaria. RSC Chem Biol. 2022 Mar 2;3(5):551-560. doi: 10.1039/d1cb00251a), allergy [37] (Anderson RJ, Tang CW, Daniels NJ, Compton BJ, Hayman CM, Johnston KA, Knight DA, Gasser O, Poyntz HC, Ferguson PM, Larsen DS, Ronchese F, Painter GF, Hermans IF. A self-adjuvanting vaccine induces cytotoxic T lymphocytes that suppress allergy. Nat Chem Biol. 2014 Nov;10(11):943-9. doi: 10.1038/nchembio.1640.), hematopoietic [38] (Grasso C, Field CS, Tang CW, Ferguson PM, J Compton B, Anderson RJ, Painter GF, Weinkove R, F Hermans I, Berridge MV. Vaccines adjuvanted with an NKT cell agonist induce effective T-cell responses in models of CNS lymphoma. Immunotherapy. 2020 Apr;12(6):395-406. doi: 10.2217/imt-2019-0134.) and solid tumors [39]. [40] (Burn OK, Farrand K, Pritchard T, Draper S, Tang CW, Mooney AH, Schmidt AJ, Yang SH, Williams GM, Brimble MA, Kandasamy M, Marshall AJ, Clarke K, Painter GF, Hermans IF, Weinkove R. Glycolipid-peptide conjugate vaccines elicit CD8+ T-cell responses and prevent breast cancer metastasis. Clin Transl Immunology. 2022 Jul 3;11(7):e1401. doi: 10.1002/cti2.1401; Speir M, Authier-Hall A, Brooks CR, Farrand KJ, Compton BJ, Anderson RJ, Heiser A, Osmond TL, Tang CW, Berzofsky JA, Terabe M, Painter GF, Hermans IF, Weinkove R. Glycolipid-peptide conjugate vaccines enhance CD8+ T cell responses against human viral proteins. Sci Rep. 2017 Oct 27;7(1):14273. doi: 10.1038/s41598-017-14690-5.). Moreover, α-GalCer -peptide conjugate administration further demonstrated the efficacy of the simultaneous co-delivery of iNKT and T cell antigens compared to the use of free α-GalCer, suggesting the importance of releasing both components within the same antigen-presenting cell, for the iNKT-mediated stimulation of antigen specific T cells.
In the present paper we report a statistically significant induction of antigen specific CD8+ T lymphocytes measured after phage injection in vivo in HHK mice (around 1.5% of NY-ESO-1/dextramer positive cells). We want to emphasize that this response is obtained in vivo in transgenic animals that possess a very low percentage of CD8+ T lymphocytes, with considerable interindividual differences, as stated by us in the discussion section (lanes 410-412 and ref [42]). Indeed, it would be interesting to compare in vivo in HHK mice the CD8 T cell response to bacteriophage co-delivering a TAA and a-GalCer respect to the peptide-glycolipid conjugate, and we hope to be able to address this issue in a future work.
However, the aim of the present work is to demonstrate the effectiveness of the bacteriophage delivery system when decorated with the α-GalCer adjuvant, in comparison to the unconjugated one and we only use the peptide antigen as positive control (mixed with CpG and IFA adjuvants in figure 3). The use of 1:1 conjugate of antigenic peptide to α-GalCer suggested from the reviewer as gold standard would represent a direct comparison among different delivery systems, comparison that is not the aim of this manuscript. Here we focus our research on filamentous bacteriophage as antigenic carrier because the phage is a natural nanoparticle with some interesting immunological properties compared to synthetic nanoparticles. For instance, bacteriophage possesses an intrinsic adjuvanticity that allows its use in the absence of exogenous adjuvants, as highlighted in the text (refs [19][20] and new refs [21][22]: Hashiguchi S, Yamaguchi Y, Takeuchi O, Akira S, Sugimura K. Immunological basis of M13 phage vaccine: Regulation under MyD88 and TLR9 signaling. Biochem Biophys Res Commun. 2010 Nov 5;402(1):19-22. doi: 10.1016/j.bbrc.2010.09.094. Epub 2010 Sep 26. PMID: 20875795; Chen F, Jiang R, Wang Y, Zhu M, Zhang X, Dong S, Shi H, Wang L. Recombinant Phage Elicits Protective Immune Response against Systemic S. globosa Infection in Mouse Model. Sci Rep. 2017 Feb 6;7:42024. doi: 10.1038/srep42024. PMID: 28165018; PMCID: PMC5292741).
In addition, bacteriophages stability in storage and transport, low cost of production and proven safety in humans together with the possibility of targeting specific cell subsets by co-displaying scFVs or receptor ligands, may make these nanoparticles extremely useful for immune therapy.
Secondly, the authors claim that they have conjugated aGalCer to the phage, which normally means that there is chemical conjugation (e.g. https://link.springer.com/protocol/10.1385/1-59259-679-7:255) In this case, the authors claim that binding is conjugation. The term as used here is misleading.
We thank the reviewer for rising this point. We reckon that the binding of lipid to phage is a not covalent bond and we have now replaced the word “conjugated” throughout the manuscript with the words “decorated” or “bound”.
The authors state that the major coat protein g8p of fd Is hydrophobic and this is why aGalCer binds to the phage. However, in the phage structure the hydrophobic portion of the g8p is not exposed at the surface (Marvin 2006 JMB and other publications), and the phage virion surface is hydrophilic. The explanation given in the manuscript must be removed as it is factually incorrect.
We thank the reviewer for the clarification. The sentence was removed accordingly.
Authors provide no proof that aGalCer is actually bound to the phage. The authors should, using aGalCer conjugates to fluorescent dyes, show that, when phage are resolved by native phage electrophoresis, the bands are labelled by the dye. Another way to demonstrate association of aGalCer to the phage is preparative agarose gel electrophoresis, extraction and mass spectroscopy.
We thank the reviewer for giving us the opportunity to better clarify this point. In a previous work we have extensively demonstrated the binding of α-GalCer to the phage surface using quantitative mass analysis (Sartorius et al, Front Immunol,2018). As described in Sartorius et al, after lipid binding, the bacteriophage/ α-GalCer was subjected to ultracentrifugation on a cesium chloride gradient in order to purify the bacteriophage/aGalCer and to remove the α-GalCer not bound to the phage. Bacteriophage/ α-GalCer was then dialyzed to further remove free α-GalCer traces. Furthermore, the bacteriophage was analyzed by Quantitative LC –MS/MS analysis to quantify the amount of lipid effectively bound to the phage. The lipid was released from phage by solvent extraction and the free glycosphingolipid was measured by a UPLC-MS/MS method. We would also like to emphasize that phage preparations used in this work were prepared accordingly to the same protocol described in Sartorius et al. We have now amended the manuscript reporting the following sentence (lanes 156-157):
The presence of α-GalCer in phage preparation was analyzed by UPLC-MS/MS method as reported in Sartorius et al, Front. Immunol, 2018.
In addition, in this work we clearly demonstrated that the iNKT response to bacteriophage/α-GalCer is higher than the one obtained using bacteriophage mixed to free α-GalCer, suggesting that the α-GalCer is bound to the phage particle and not free.
Round 2
Reviewer 2 Report
It is nice to see that the authors have referred to the relevant immunology literature. However, it is done so only in discussion. High induction of the T-cell responses by the peptide-aGalCer chemical conjugates should also be mentioned in the introduction. Having these findings in mind, potential advantages of filamentous phage-based vaccine over a chemical conjugate should be included in the introduction.
Discussion:
Line 393
The use of synthetic peptides bound to immunostimulating lipids, such as ….
Replace with
The use of synthetic peptides chemically conjugated to immunostimulating lipids, such as….
Author Response
It is nice to see that the authors have referred to the relevant immunology literature. However, it is done so only in discussion. High induction of the T-cell responses by the peptide-aGalCer chemical conjugates should also be mentioned in the introduction. Having these findings in mind, potential advantages of filamentous phage-based vaccine over a chemical conjugate should be included in the introduction.
We thank the reviewer for the suggestion. The use of the peptide-a-GalCer chemical conjugates is now reported in the introduction (lines 67-70):
For instance, the chemical conjugation of α-GalCer to antigenic peptides has shown the induction of an efficient stimulatory activity by iNKT cells, with the enhancement of cytotoxic T cell responses and therapeutic anti-tumor effects [19] (Anderson, R.J.; Compton, B.J.; Tang, C.W.; Authier-Hall, A.; Hayman, C.M.; Swinerd, G.W.; Kowalczyk, R.; Harris, P.; Brimble, M.A.; Larsen, D.S.; et al. NKT Cell-Dependent Glycolipid-Peptide Vaccines with Potent Anti-Tumour Activity. Chem. Sci. 2015, 6, 5120–5127, doi:10.1039/c4sc03599b.) [20] (Burn, O.K.; Farrand, K.; Pritchard, T.; Draper, S.; Tang, C. wen; Mooney, A.H.; Schmidt, A.J.; Yang, S.H.; Williams, G.M.; Brimble, M.A.; et al. Glycolipid-Peptide Conjugate Vaccines Elicit CD8+ T-Cell Responses and Prevent Breast Cancer Metastasis. Clin. Transl. Immunol. 2022, 11, doi:10.1002/cti2.1401.
In addition, we also discuss the potential advantages of filamentous phage-based vaccine over a chemical conjugate (lines 84-94):
Phage-based vaccine simultaneously delivering α-GalCer and antigenic peptides offers many advantages compared to α-GalCer-peptide chemical conjugates: phages possess an intrinsic adjuvanticity that allows their use in the absence of exogenous adjuvants, promoting DC maturation and potentially resulting in synergistic effects. In addition, the expression of antigenic peptides in multiple copies on a single phage particle permits the strong multivalent binding of peptides to the APCs, increases the peptide stability and optimizes the biodistribution and pharmacokinetics [27][28][29] (González-Mora, A.; Hernández-Pérez, J.; Iqbal, H.M.N.; Rito-Palomares, M.; Benavides, J. Bacteriophage-Based Vaccines: A Potent Approach for Antigen Delivery. Vaccines 2020; Stern, Z.; Stylianou, D.C.; Kostrikis, L.G. The Development of Inovirus-Associated Vector Vaccines Using Phage-Display Technologies. Expert Rev. Vaccines 2019.;Zou, J.; Dickerson, M.T.; Owen, N.K.; Landon, L.A.; Deutscher, S.L. Biodistribution of Filamentous Phage Peptide Libraries in Mice. Mol. Biol. Rep. 2004, doi:10.1023/B:MOLE.0000031459.14448.af.).Moreover, α-GalCer bacteriophages can be further modified, to generate multivalent phages exposing different types of peptides [30] (Berardinis, P.; Sartorius, R.; Caivano, A.; Mascolo, D.; Domingo, G.; Pozzo, G.; Gaubin, M.; Perham, R.; Piatier-Tonneau, D.; Guardiola, J. Use of Fusion Proteins and Procaryotic Display Systems for Delivery of HIV-1 Antigens: Development of Novel Vaccines for HIV-1 Infection. Curr. HIV Res. 2005, doi:10.2174/1570162033485168.), or offering the possibility of targeted delivery to a specific tissue or cell subsets by co-displaying antibody fragments or receptor ligands on phage surface, maximizing the specific uptake [23][31][32] (Sartorius, R.; D’Apice, L.; Trovato, M.; Cuccaro, F.; Costa, V.; De Leo, M.G.; Marzullo, V.M.; Biondo, C.; D’Auria, S.; De Matteis, M.A.; et al. Antigen Delivery by Filamentous Bacteriophage Fd Displaying an Anti‐DEC‐205 Single‐chain Variable Fragment Confers Adjuvanticity by Triggering a TLR 9‐mediated Immune Response. EMBO Mol. Med. 2015, doi:10.15252/emmm.201404525; Murgas, P.; Bustamante, N.; Araya, N.; Cruz-Gómez, S.; Durán, E.; Gaete, D.; Oyarce, C.; López, E.; Herrada, A.A.; Ferreira, N.; et al. A Filamentous Bacteriophage Targeted to Carcinoembryonic Antigen Induces Tumor Regression in Mouse Models of Colorectal Cancer. Cancer Immunol. Immunother. 2018, doi:10.1007/s00262-017-2076-x.; Safaei Ghaderi, S.; Riazi-Rad, F.; Safaie Qamsari, E.; Bagheri, S.; Rahimi-Jamnani, F.; Sharifzadeh, Z. Development of a Human Phage Display-Derived Anti-Pd-1 Scfv Antibody: An Attractive Tool for Immune Checkpoint Therapy. SSRN Electron. J. 2022, doi:10.2139/ssrn.4025854).
Discussion:
Line 393
The use of synthetic peptides bound to immunostimulating lipids, such as ….
Replace with
The use of synthetic peptides chemically conjugated to immunostimulating lipids, such as….
We thank the reviewer for pointing it out. The sentence was modified accordingly (now line 406).